# Could Prior COVID-19 Affect the Neutralizing Antibody after the Third BNT162b2 Booster Dose: A Longitudinal Study

**DOI:** 10.3390/vaccines11030560

**Published:** 2023-03-01

**Authors:** Mustafa Genco Erdem, Ozge Unlu, Suleyman Buber, Mehmet Demirci, Bekir Sami Kocazeybek

**Affiliations:** 1Department of Internal Medicine, Faculty of Medicine, Beykent University, İstanbul 34398, Türkiye; 2Department of Medical Microbiology, Faculty of Medicine, Istanbul Atlas University, İstanbul 34403, Türkiye; 3Department of Medical Biochemistry, Medicalpark Gaziosmanpasa Hospital, Faculty of Medicine, Istinye University, İstanbul 34240, Türkiye; 4Department of Medical Microbiology, Faculty of Medicine, Kirklareli University, Kırklareli 39100, Türkiye; 5Department of Medical Microbiology, Cerrahpaşa Faculty of Medicine, Istanbul University-Cerrahpaşa, İstanbul 34098, Türkiye

**Keywords:** surrogate neutralizing antibody, anti-S-RBD IgG titers, third dose, BNT162b2

## Abstract

Vaccination is an essential public health measure for preventing the spread of illness during this continuing COVID-19 epidemic. The immune response developed by the host or the continuation of the immunological response caused by vaccination is crucial since it might alter the epidemic’s prognosis. In our study, we aimed to determine the titers of anti-S-RBD antibody and surrogate neutralizing antibody (snAb) formed before and after the third dose of the BNT162b2 vaccination (on the 15th, 60th, and 90th days) in healthy adults who did not have any comorbidity either with or without prior SARS-CoV-2 infection. In this longitudinal prospective study, 300 healthy persons were randomly included between January and February 2022, following two doses of BNT162b2 immunization and before a third dosage. Blood was drawn from the peripheral veins. SARS-CoV-2 NCP IgG and anti-S-RBD IgG levels were detected by the CMIA method, and a surrogate neutralizing antibody was seen by the ELISA method. Our study included 154 (51.3%) female and 146 (48.7%) male (total 300) participants. The participants’ median age was 32.5 (IQR:24–38). It was discovered that 208 individuals (69.3%) had never been infected with SARS-CoV-2, whereas 92 participants (30.7%) had SARS-CoV-2 infections in the past. Anti-S-RBD IgG and nAb IH% levels increased 5.94- and 1.26-fold on day 15, 3.63- and 1.22-fold on day 60, and 2.33- and 1.26-fold on day 90 after the third BNT162b2 vaccine dosage compared to pre-vaccination values (Day 0). In addition, the decrease in anti-S-RBD IgG levels on the 60th and 90th days was significantly different in the group without prior SARS-CoV-2 infection compared to the group with past SARS-CoV-2 infection (*p <* 0.05). In conclusion, it was observed that prior SARS-CoV-2 infection and the third BNT162b2 vaccine dose led to a lower decrease in both nAb and anti-S-RBD IgG levels. To evaluate the vaccine’s effectiveness and update immunization programs, however, it is necessary to perform multicenter, longer-term, and comprehensive investigations on healthy individuals without immune response issues, as there are still circulating variants.

## 1. Introduction

COVID-19 is a highly contagious virus produced by severe acute respiratory syndrome Coronavirus-2 (SARS-CoV-2), which arose 101 years after the influenza pandemic. The COVID-19 pandemic has had disastrous impacts on a global scale which continue [1,2]. Therefore, vaccination is an essential public health approach for preventing the spread of illness. During the COVID-19 pandemic, the Pfizer-BioNTech (BNT162b2) and Moderna (mRNA-1273) vaccines, which were produced using messenger RNA (mRNA) technology, were widely employed as a form of disease prevention [3]. Although studies examine the immunological response of individuals to the BNT162b2 vaccination, mutations occur during the proliferation of SARS-CoV-2, resulting in the formation of SARS-CoV-2 variants that can modify the immune response. Thus, monitoring the immune response that develops after SARS-CoV-2 infection and the immune response caused by vaccination is crucial since they may alter the epidemic’s prognosis [4,5]. Protection against SARS-CoV-2 infections depends on the longevity of the vaccine-induced immune response within the community. For immunization efforts to provide longer-term protection, investigations monitoring the immune response to COVID-19 following vaccine doses are required [6]. Anti-spike protein receptor-binding domain (S-RBD) antibodies are required to demonstrate personal protection against SARS-CoV-2 infection. In addition, neutralization antibodies (nAbs) attach to viral proteins and are essential in blocking viral entrance during host–virus contact. Surrogate neutralizing antibody assays are immunological tests that can identify SARS-CoV-2 neutralizing antibody levels in the host without using a live virus [7,8]. This investigation aimed to determine the titers of anti-S-RBD antibody and surrogate neutralizing antibody (snAb) generated before and after the third BNT162b2 vaccination (on the 15th, 60th, and 90th days) in healthy persons with or without past SARS-CoV-2 infection and without comorbidities.

## 2. Materials and Methods

### 2.1. Study Design and Samples Collection

In this prospective longitudinal study, 300 healthy persons were recruited to participate at random after two doses of BNT162b2 immunization and prior to a third dose administered between January and February 2022. All individuals were required to complete an extra comorbidity questionnaire. Age, gender, and comorbidities were used to conduct a complete evaluation. The exclusion criteria were as follows: (I) age 18 or >50, (II) a history of comorbidities, and (III) a current infection. This research included participants for the third vaccine dosage with a median of 151 (IQR [Inter Quartile Range 25–75 percentiles]: 108–161) days following the second vaccination dose. Before the study, all participants provided written and informed consent, and ethical approval was granted by the Ethics Committee of the Kirklareli University Faculty of Medicine (approval number: E-37844677-199-40832) and the Republic of Turkey Ministry of Health General Directorate of Health Services Scientific Research Studies Commission (approval number: 2021-11-22T21_04_43). During each blood draw, a Panbio COVID-19 (Abbott, IL, USA) fast antigen test kit was used according to the manufacturer’s instructions to detect active SARS-CoV-2 infection in the subjects. Blood samples were taken just before the third vaccination (0th). Fifteen days, sixty days, and ninety days following the third immunization dose, blood samples were taken for follow-up. On the day of collection, all peripheral blood samples were transferred to the laboratory and processed.

### 2.2. Antibody Testing

Peripheral blood samples were centrifuged at 400× *g* for 10 min at room temperature to extract serum samples. The Abbott Architect SARS-CoV-2 immunoglobulin G (IgG) test (Abbott, IL, USA), which semi-quantitatively identifies IgG antibodies against the Nucleocapsid protein (NCP) of SARS-CoV-2, was used to detect contact with SARS-CoV-2 in the participants. The participants with a concentration above the median (IQR) of 2.03 signal-to-cutoff (S/Co) ratios were considered previous SARS-CoV-2 infection. Abbott Architect SARS-CoV-2 IgG II Quant test (Abbott, IL, USA) was used according to the manufacturer’s instructions to detect IgG antibodies against the receptor-binding region (RBD) of the spike protein S1 subunit of SARS-CoV-2. These data can be converted to “Binding Antibody Unit per milliliter (BAU/mL)” according to the WHO’s International Standard (multiplied by the correlation coefficient of 0.142) [9].

To detect possibly neutralizing or probably neutralizing antibodies (nAb) against SARS-CoV-2, the SARS-CoV-2 NeutraLISA test (Euroimmun, Lübeck, Germany) was utilized as a surrogate neutralization antibody (snAb) assay according to the manufacturer’s instructions. This test determined the presence of an antibody that prevents RBD from binding to ACE2. Results were evaluated as neutralizing antibody (nAb) percent inhibition (IH%). Tests with a nAb IH% ≥35% were regarded as positive, and tests with a nAb IH% <20% were deemed negative. According to the manufacturer’s instructions, nAb IH% between 20% and 35% was deemed borderline [10,11].

### 2.3. Statistical Analysis

The software version 20 of IBM SPSS was utilized. The data are shown as a median and interquartile range (IQR 25–75 percentiles), as well as a number (N) and a percentage (%). Comparisons between groups were analyzed using Mann–Whitney U tests. In all studies, a *p* value < 0.05 was considered significant. In addition, Spearman correlation analysis was performed to identify correlations. For Spearman’s correlation value (rs), rs < 0.25 was considered not statistically correlated, rs = 0.25–0.5 was considered a weak correlation, rs = 0.5–0.75 was considered a moderate correlation, rs = 0.76–0.85 was considered a strong correlation, and rs > 0.85 was considered a very strong correlation.

## 3. Results

Following two doses of BNT162b2, 154 (51.3%) of the 300 subjects admitted for the third BNT162b2 vaccine dose and included in our study were female, while 146 (48.7%) were male. In addition, the participants’ median age was 32.5 (IQR: 24–38) (Table 1).

Anti-S-RBD IgG levels increased 5.94-fold on day 15, 3.63-fold on day 60, and 2.33-fold on day 90 after the third BNT162b2 vaccine dosage compared to pre-vaccination values (Day 0). As comparison to pre-vaccination levels, the subjects’ nAb IH% levels increased 1.26-fold on day 15, 1.22-fold on day 60, and 1.13-fold on day 90 following the third BNT162b2 vaccine dose (day 0). (Figure 1, Figure 2 and Figure 3). Anti-S-RBD IgG titer and nAb IH% levels did not differ statistically significantly between males and females (*p >* 0.05).

Anti-SARS-CoV-2 NCP IgG levels were measured before the third vaccination dose in a total of 300 study participants; 208 (69.3%) participants with anti-SARS-CoV-2 NCP IgG levels of 2.03 S/Co or less had never had a SARS-CoV-2 infection, whereas 92 (30.7%) participants with anti-SARS-CoV-2 NCP IgG levels above 2.03 S/Co had a previous SARS-CoV infection (Table 2). In the groups without and with past SARS-CoV-2 infection, anti-S-RBD IgG levels rose 4.96- and 7.52-fold on day 15, 2.93- and 5.05-fold on day 60, and 1.52- and 4.20-fold on day 90 compared to pre-vaccination (Day 0). At day 60 and day 90, anti-S-RBD IgG levels decreased significantly in both groups with and without prior SARS-CoV-2 infection. In the groups without and with past SARS-CoV-2 infection, nAb IH% levels rose 1.29- and 1.24-fold on day 15, 1.26- and 1.19-fold on day 60, and 1.13- and 1.12-fold on day 90 compared to pre-vaccination (Day 0), respectively. nAb IH% levels recorded on day 0 and day 90 were significantly lower in the group without prior SARS-CoV-2 infection than in the group with prior SARS-CoV-2 infection (*p <* 0.05) (Table 2).

Anti-SARS-CoV-2 NCP IgG levels were analyzed before the third vaccination dose in 154 (51.3%) women who participated in the study. Ninety-eight (63.6%) of the 154 women with anti-SARS-CoV-2 NCP IgG levels of 2.03 S/Co or less had never been infected with SARS-CoV-2. Nevertheless, 56 of the 154 women (36.4%) with anti-SARS-CoV-2 NCP IgG levels more than 2.03 S/Co had been infected with SARS-CoV-2. In women without and with past SARS-CoV-2 infection, anti-S-RBD IgG levels rose 17.88- and 5.25-fold on the 15th day, 10.92- and 3.46-fold on the 60th day, and 8.94- and 2.60-fold on the 90th day following the third BNT162b2 vaccine dose, compared to the pre-vaccination values (Day 0). In groups without and with past SARS-CoV-2 infection, nAb IH% levels rose 1.28- and 1.23-fold on day 15, 1.18- and 1.18-fold on day 60, and 1.11- and 1.11-fold on day 90, compared to pre-vaccination (day 0). Anti-S-RBD IgG and nAb IH% levels did not differ significantly between groups of women with and without prior SARS-CoV-2 infection (*p* > 0.05) (Table 3).

When anti-SARS-CoV-2 NCP IgG levels were analyzed before the third vaccination dose in 146 (48.7%) male participants, it was shown that 110 of the 146 (75.3%) males with anti-SARS-CoV-2 NCP IgG levels of 2.03 S/Co or less had never been infected with SARS-CoV-2. Nevertheless, anti-SARS-CoV-2 NCP IgG levels were over 2.03 S/Co in 36 of 146 (24.7%) males previously infected with SARS-CoV-2. In males without and with past SARS-CoV-2 infection, anti-S-RBD IgG titer rose 3.20- and 17.46-fold on the 15th day, 2.57- and 12.54-fold on the 60th day, and 1.02- and 8.68-fold on the 90th day, compared to pre-vaccination levels (day 0). In addition, nAb IH% levels increased 1.31- and 1.28-fold on day 15, 1.28- and 1.20-fold on day 60, and 1.18- and 1.11-fold on day 90, compared to pre-vaccination (Day 0), in the group without and with prior SARS-CoV-2 infection, respectively. Furthermore, nAb IH% levels rose 1.31- and 1.28-fold on day 15, 1.28- and 1.20-fold on day 60, and 1.18- and 1.11-fold on day 90, relative to pre-vaccination (Day 0) in the groups without and with detected SARS-CoV-2 exposure, respectively. The decrease in anti-S-RBD IgG titers on days 60 and 90 were statistically significant in both groups (*p <* 0.05): those without and those with past SARS-CoV-2 infection. nAb IH% levels were statistically significant only before the third vaccination dose (day 0) in the group without prior SARS-CoV-2 infection and the group with past infection (*p <* 0.05). After the third vaccination (day 15), nAb IH% levels in the infected and uninfected groups increased to a similar level (*p* > 0.05) (Table 4).

While there was a moderate positive association between NCP IgG titer and nAb IH% level before the third vaccination dosage (day 0) (rs:0.511, *p <* 0.05), there was only a weak positive correlation on the 15th, 60th, and 90th days following the third vaccination dose. (rs:0.399, *p <* 0.05), and a weak positive correlation was seen between anti-S-RBD IgG titers and nAb IH% level at all time periods (rs:0.336, *p <* 0.05).

## 4. Discussion

Monitoring SARS-CoV-2 antibodies in the host is essential for determining the significance of these antibodies in avoiding illness and revising vaccination policy [11]. Therefore, determining the changes in anti-S-RBD antibody and surrogate neutralizing antibody (snAb) levels in healthy persons before and after the third BNT162b2 vaccination (on the 15th, 60th, and 90th days) was the focus of our investigation.

In longitudinal investigations examining the anti-S-RBD IgG level of the third BNT162b2 vaccination dosage, Lo Sasso et al. found that women have a greater baseline antibody level than men. They also found that anti-S-RBD antibody levels fell to a steady state after four months, and anti-S-RBD IgG levels were independent of age, gender, vaccine doses, and baseline antibody titer [12]. Women without prior SARS-CoV-2 infection did not have a greater baseline anti-S-RBD IgG level than men without prior SARS-CoV-2 infection, according to the findings of this study. However, in cases of prior SARS-CoV-2 infection, baseline antibody levels were higher in women than in males. Antibody levels reverted to baseline at the end of the third month (90th day) in men and women without prior SARS-CoV-2 infection. Anti-S-RBD IgG levels were higher than baseline at the end of the third month (90th day) in both genders with past SARS-CoV-2 infection. Eliakim-Raz et al. reported that 10–19 days following the third vaccination dosage, anti-S-RBD IgG levels increased to 25,468 AU/mL in those aged 60 and older [13]. In our study, on the 15th day following the third vaccination, anti-S-RBD IgG levels climbed to around 32,000 AU/mL in all individuals in the healthy group younger than 50. Anti-S-RBD IgG levels were approximately 38,000 AU/mL in the group with prior SARS-CoV-2 infection compared to 28,000 AU/mL in the group without prior infection. Romero-Ibarguengoitia et al. found that 1–7 days after the third vaccination dose, anti-S-RBD IgG levels were significantly different between individuals with and without prior SARS-CoV-2 infection between days 21 and 28, but this statistical difference faded between days 21 and 28 [14]. Anti-S-RBD IgG levels on days 0 and 15 in patients with or without prior SARS-CoV-2 infection did not differ statistically, according to the results of our study.

After the third dosage of the BNT162b2 vaccine, Kontopoulou et al. discovered anti-S-RBD IgG titers of around 20,000 AU/mL. They reported that they did not detect any difference between men and women. Likewise, this study found no difference between patients with or without prior SARS-CoV-2 infection [15]. Similarly, our study observed no difference in anti-S-RBD IgG levels between men and women, as well as between those with and without prior SARS-CoV-2 infection. When longitudinal studies examining the impact of the third BNT162b2 vaccine dosage on neutralizing antibody levels are assessed, Falsey et al. reported that the wild-type virus provides a five-fold greater level of neutralizing antibodies one month after vaccination [16]. Our study found that surrogate neutralizing antibody levels increased by 28% and 31% in women and men, respectively, on the 15th day, which was slightly lower than in the group without prior SARS-CoV-2 infection, but the difference was not statistically significant. Matula et al. observed that the percentage of neutralizing antibodies rose from 65% to 98% fourteen days following the third dose of BNT162b2 vaccination [17]. As a result of our study, we determined that the nAb IH% increased from around 74% to 93%. There was no statistically significant difference between the group with or without prior SARS-CoV-2 infection regarding nAb IH% rise.

Cassaniti et al. observed that the nAb level in kidney transplant recipients decreased from 1/20 on the 21st day to 1/10 three months after the third BNT162b2 vaccination dose [18]. Our study showed that the percentage of nAb IH on day 90 dropped by roughly 10% compared to day 15. The difference in immunological response between our study and that of Cassaniti et al. [18] may be because our study cohort consisted of healthy individuals.

It is known that, after vaccination, high-potency matured antibodies targeting conserved SARS-CoV-2 RBD region can be produced, although the formation of nAb and anti-S-RBD antibody titers can be decreased in the host against novel variants such as the Omicron. Spike-specific memory B cells ensure the rapid and efficient generation of antibodies. Since natural infection increases the formation of memory B cells, the host immune response may become more robust [19]. It has been demonstrated to contribute to both humoral and cellular responses against SARS-CoV-2 in polyfunctional memory CD8+T cells and memory CD4+T cells, as well as in RBD-specific memory B cells [20]. In order to comprehend the differential immune response mechanism, it may be required to monitor the T cell response status as well as the antibody response.

Our study’s limitation is that more frequent and prolonged follow-ups are not carried out in different centers. This restriction has arisen due to the fact that healthy individuals do not want to visit hospitals because of the disease risk. Furthermore, individuals above the age of 50 were excluded from our study since they were less physically active and had more comorbidities than those between the ages of 18 and 50, which might introduce bias into the findings. In addition, they do not know the variant type or timing of SARS-CoV-2 variants previously encountered.

## 5. Conclusions

In conclusion, our study showed that in healthy adults, anti-S-RBD IgG titers increased approximately six-fold on the 15th day and decreased three-fold on the 90th day compared to the 15th day. In addition, nAb IH% increased by 26% on the 15th day and decreased by two-fold on the 90th day compared to the 15th day. The presence of a prior SARS-CoV-2 infection, as well as the third dose of the BNT162b2 vaccination, was observed to reduce nAb and anti-S-RBD IgG levels to a lesser extent. A prior SARS-CoV-2 infection in men or women did not change nAb levels significantly, and a prior SARS-CoV-2 infection prolonged anti-S-RBD titers but did not affect nAb levels. As multiple variations are still circulating, monitoring vaccination effectiveness and updating immunization programs requires multicenter, longer-term, and comprehensive studies in healthy individuals without immune response issues.

## Figures and Tables

**Figure 1 vaccines-11-00560-f001:**
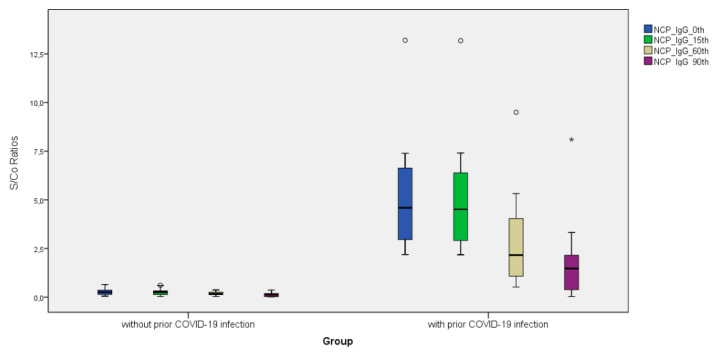
Distribution of SARS-CoV-2 NCP IgG titers (S/Co ratios) before and after the third dose of BNT162b2 vaccine (on the 15th, 60th, and 90th days) with and without prior SARS-CoV-2 infection.

**Figure 2 vaccines-11-00560-f002:**
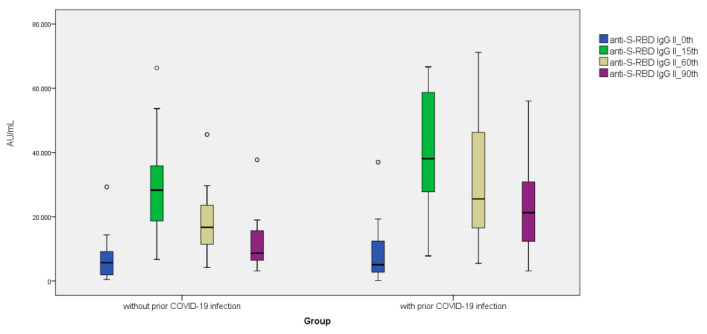
Distribution of anti-S-RBD antibody titers (AU/mL) before and after the third dose of BNT162b2 vaccine (on the 15th, 60th, and 90th days) with and without prior SARS-CoV-2 infection.

**Figure 3 vaccines-11-00560-f003:**
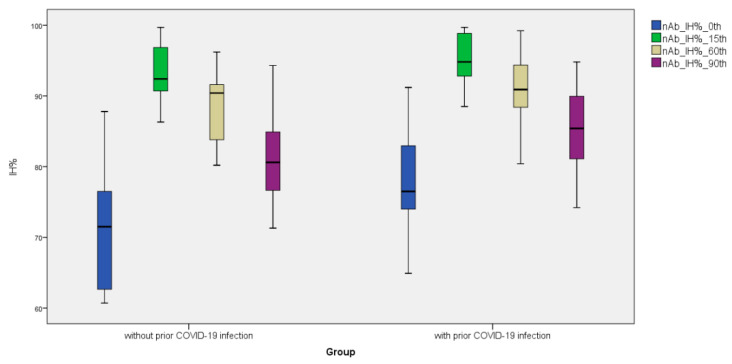
Distribution of surrogate neutralizing antibody (snAb) titers (IH%) formed before and after the third dose of BNT162b2 vaccine (on the 15th, 60th, and 90th days) with and without prior SARS-CoV-2 infection.

**Table 1 vaccines-11-00560-t001:** Demographic data and distributions of the levels of anti-SARS-CoV-2 NCP IgG, anti-S-RBD IgG, and snAb IH% of all participants before the third vaccine dose (day 0) and after (days 15, 60 and 90).

	All Participants (*n*: 300)			Female (*n*: 154)			Male (*n*: 146)			Female vs. Male
	Median	IQR25	IQR75	Median	IQR25	IQR75	Median	IQR25	IQR75	*p*
Gender (F/M) *n* (%)	154 (51.3%)/146 (48.7%)									
Age	32.50	24.00	38.00	32	23.5	45	33	24.75	37	0.869
NCP_IgG_0th (S/Co)	0.65	0.25	4.05	2.59	0.38	4.27	0.29	0.14	4.22	0.062
NCP_IgG_15th (S/Co)	0.59	0.23	3.99	2.54	0.38	4.21	0.27	0.16	4.16	0.062
NCP_IgG_60th (S/Co)	0.28	0.15	2.12	0.67	0.18	2.57	0.25	0.14	1.87	0.42
NCP_IgG_90th (S/Co)	0.24	0.06	1.26	0.25	0.02	1.57	0.21	0.13	1.00	0.425
anti-S-RBD IgG II_0th (AU/mL)	5380.45	2332.43	12,442.50	4218.35	530.40	11,984.03	6095.85	3273.30	14,375.30	0.236
anti-S-RBD IgG II_15th (AU/mL)	31,946.60	19,739.50	50,076.30	32,816.60	27,749.20	53,666.40	24,530.00	16,340.90	37,576.43	0.18
anti-S-RBD IgG II_60th (AU/mL)	19,538.70	13,203.05	30,467.30	19,575.40	17,438.60	32,891.60	16,712.10	11,651.23	27,105.03	0.285
anti-S-RBD IgG II_90th (AU/mL)	12,546.50	6604.80	21,720.33	16,027.70	11,419.90	23,096.50	10,364.20	6604.80	18,825.73	0.08
snAb 0th (IH%)	74.15	66.80	81.25	75.25	67.53	81.25	73.95	66.20	81.50	0.985
snAb 15th (IH%)	93.70	91.35	97.83	93.20	91.13	96.78	94.40	91.28	98.43	0.426
snAb 60th (IH%)	90.80	84.58	92.78	90.25	84.40	93.73	91.10	85.68	92.48	0.62
snAb 90th (IH%)	83.85	78.03	89.40	83.15	77.10	89.58	84.15	78.50	88.28	0.869

**Table 2 vaccines-11-00560-t002:** Distribution of anti-SARS-CoV-2 NCP IgG, anti-S-RBD IgG and snAb IH% levels in all participants before (day 0) and after (days 15, 60, and 90) the third vaccine dose.

All Participants	without Prior SARS-CoV-2 Infection (*n*:208)			with Prior SARS-CoV-2 Infection (*n*:92)			*p*
	Median	IQR25	IQR75	Median	IQR25	IQR75	
Gender (F/M)	98/110			56/36			0.251
Age	32.00	23.00	37.00	33.00	26.00	44.00	0.558
NCP_IgG_0th (S/Co)	0.25	0.14	0.38	4.59	2.81	6.69	0.000
NCP_IgG_15th (S/Co)	0.26	0.13	0.36	4.51	2.72	6.39	0.000
NCP_IgG_60th (S/Co)	0.18	0.12	0.25	2.16	1.01	4.23	0.000
NCP_IgG_90th (S/Co)	0.13	0.02	0.18	1.47	0.36	2.18	0.000
anti-S-RBD IgG II_0th (AU/mL)	5700.30	763.50	11,831.20	5060.60	2340.70	12,442.50	0.477
anti-S-RBD IgG II_15th (AU/mL)	28,281.10	16,621.30	36,075.00	38,076.90	27,749.20	61,720.00	0.052
anti-S-RBD IgG II_60th (AU/mL)	16,712.10	10,041.90	23,580.10	25,550.00	14,772.80	52,584.00	0.019
anti-S-RBD IgG II_90th (AU/mL)	8662.70	6262.30	16,027.70	21,261.60	12,162.50	34,034.50	0.001
snAb 0th (IH%)	71.50	61.80	76.50	76.50	73.80	84.30	0.003
snAb 15th (IH%)	92.40	90.30	97.20	94.80	92.80	99.10	0.076
snAb 60th (IH%)	90.40	83.50	91.70	90.90	86.70	94.50	0.113
snAb 90th (IH%)	80.60	76.50	85.40	85.40	79.80	90.10	0.047

**Table 3 vaccines-11-00560-t003:** Distributions of anti-SARS-CoV-2 NCP IgG, anti-S-RBD IgG, and snAb IH% levels among women with and without prior SARS-CoV-2 infection, before (day 0) and after (days 15, 60 and 90) the third vaccine dose.

Female Participants	without Prior SARS-CoV-2 Infection (*n*:98)			with Prior SARS-CoV-2 Infection (*n*:56)			*p*
	Median	IQR25	IQR75	Median	IQR25	IQR75	
Age	32.00	24.00	37.00	36.00	22.00	45.00	0.570
NCP_IgG_0th (S/Co)	0.38	0.12	0.60	3.87	2.79	6.85	0.000
NCP_IgG_15th (S/Co)	0.38	0.12	0.55	3.81	2.72	6.79	0.000
NCP_IgG_60th (S/Co)	0.18	0.07	0.27	2.03	0.89	4.23	0.000
NCP_IgG_90th (S/Co)	0.02	0.02	0.13	1.19	0.25	2.38	0.000
anti-S-RBD IgG II_0th (AU/mL)	1792.10	504.45	9841.63	6511.20	2332.43	12,442.50	0.149
anti-S-RBD IgG II_15th (AU/mL)	32,040.80	20,282.23	49,154.83	34,154.40	27,749.20	61,720.00	0.470
anti-S-RBD IgG II_60th (AU/mL)	19,575.40	8739.48	29,659.20	22,526.00	17,438.60	37,814.70	0.570
anti-S-RBD IgG II_90th (AU/mL)	16,027.70	5757.43	18,995.20	17,044.20	11,419.90	25,831.00	0.606
snAb 0th (IH%)	72.35	60.90	78.98	76.50	71.45	82.13	0.080
snAb 15th (IH%)	92.35	89.85	96.50	94.05	91.70	97.75	0.382
snAb 60th (IH%)	85.45	83.43	93.00	90.60	86.60	94.13	0.142
snAb 90th (IH%)	80.15	76.58	88.13	85.00	79.43	90.10	0.150

**Table 4 vaccines-11-00560-t004:** Distributions of anti-SARS-CoV-2 NCP IgG, anti-S-RBD IgG, and snAb IH% levels among men with and without prior SARS-CoV-2 infection, before (day 0) and after (days 15, 60, and 90) the third vaccine dose.

Male Participants	without Prior SARS-CoV-2 Infection (*n*:110)			with Prior SARS-CoV-2 Infection (*n*:36)			*p*
	Median	IQR25	IQR75	Median	IQR25	IQR75	
Age	34.00	23.00	37.00	33.00	30.00	35.00	0.881
NCP_IgG_0th (S/Co)	0.18	0.14	0.29	4.59	2.85	6.63	0.000
NCP_IgG_15th (S/Co)	0.16	0.13	0.27	4.51	2.82	6.39	0.000
NCP_IgG_60th (S/Co)	0.15	0.13	0.25	3.21	1.08	4.11	0.000
NCP_IgG_90th (S/Co)	0.15	0.12	0.24	1.88	0.41	2.45	0.001
anti-S-RBD IgG II_0th (AU/mL)	6491.40	4999.30	14,375.30	3184.20	2255.35	27,255.60	0.417
anti-S-RBD IgG II_15th (AU/mL)	20,778.90	16,340.90	36,075.00	55,588.00	11,585.45	60,615.50	0.111
anti-S-RBD IgG II_60th (AU/mL)	16,712.10	10,041.90	23,580.10	39,932.80	12,299.65	64,905.10	0.044
anti-S-RBD IgG II_90th (AU/mL)	6604.80	6262.30	12,065.70	27,637.90	12,546.50	55,502.10	0.001
snAb 0th (IH%)	70.90	63.50	76.50	76.80	74.20	88.50	0.016
snAb 15th (IH%)	93.10	91.10	97.30	98.60	93.15	99.40	0.052
snAb 60th (IH%)	91.10	85.40	91.70	92.10	85.95	95.70	0.297
snAb 90th (IH%)	83.60	74.90	85.40	85.60	80.45	90.70	0.121

## Data Availability

No new data were created.

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
