# Peer review of "Could Prior COVID-19 Affect the Neutralizing Antibody after the Third BNT162b2 Booster Dose: A Longitudinal Study"

_vaccines, 2023, doi:10.3390/vaccines11030560_

Round 1

Reviewer 1 Report

The authors report the results of multiple antibody assays (anti-nucleocapsid IgG, anti-RBD IgG, and a surrogate neutralization assay reported as the percent inhibition of RBD binding to ACE2) performed at 0, 15, 60, and 90 days following a third dose of the BNT162b2 mRNA vaccine. The data was collected from a relatively large cohort of 300 individuals. Analysis focused on the impact of prior SARS-CoV-2 infection on the magnitude and duration of the vaccine-induced antibody response, as well as the impact of gender. They find that prior SARS-CoV-2 does significantly increase the amount of anti-RBD IgG at later timepoints, but that it doesn’t cause a corresponding increase in surrogate neutralizing antibody levels. The authors’ premise and data appear sound, but editing is required to ensure clarity for the reader.

1. The definition of participants with and without prior SARS-CoV-2 infection is critical to the analysis and interpretation of the reported results. The methods section taken alone is not clear about whether that is defined as having anti-nucleocapsid antibodies or an affirmative answer to a survey result; this wasn’t made fully clear until the results in lines 146-150. I then got confused again when reading lines 163-167 – is this stating that 98 of the 154 women in the study (so, 63.6% of the women) were not previously infected as defined by the presence of N antibody, and 56/154 (so, 36.4% of the women) were previously infected? For the males (lines 179-182), shouldn’t it say 110/146=75.3% of men were not infected and 36/110=24.7% of men were not previously infected? Modify the methods section to more explicitly state that only the antibody data was used to define previously infected vs. no previous infection. Also, as the results of the survey question about previous SARS-CoV-2 infection do not appear to have been used in this study, it may be easiest to remove that text and refer only to data which was subsequently used for either generating or interpreting results or for inclusion/exclusion decisions.

2. Regarding the exclusion criteria, the exclusion of individuals <18 years old is understandable due to the difficulty in obtaining truly informed consent from minors. However, please provide the rationale behind excluding those <50 years of age and/or presenting with any co-morbidities. Aren’t these the very groups at most risk from COVID-19, and wouldn’t data on this population be especially valuable? This could be included in the limitations section.

3. There are a few instances where COVID-19 and SARS-CoV-2 are used interchangeably in the text and figures. Please do a close edit to make sure that it’s always COVID-19 when describing the disease and SARS-CoV-2 when referring to the virus. For example, line 57 should refer to the immune response against SARS-CoV-2 (the virus), not COVID-19 (the disease). Similarly, the labels of Figures 1, 2, and 3 refer to COVID-19 infection when they should technically refer to SARS-CoV-2 infection.

4. Lines 45-46. I disagree somewhat is the phrasing here. Vaccination was able to help control the spread during the original and alpha variant waves, but whether it still decreases transmission in the face of the delta and omicron variants is far less certain due to the large number of breakthrough infections.

5. Change the wording regarding the surrogate neutralization assay. Clarify that this assay is measuring the presence of antibody that blocks the binding of RBD to ACE2, and that you are detecting potentially neutralizing or likely neutralizing antibodies, not neutralizing antibodies. The assay is fully valid, but make sure its clear throughout the paper that the data are for this surrogate assay, not a traditional neutralization assay by referring to the snAb that were actually measured, not nAb which was not measured.

6. Why were the %IH cutoffs of 35 for a positive and <20 for a negative chosen (lines 107-108), and how were samples that fell between those two cutoffs handled?

7. A correction for multiple comparisons needs to be applied to your statistical analysis.

8. Table 1 and Figure 2 refer to the anti-S-RBD IgG levels in terms of AU/ml. Is this correct? I assumed from the methods section that you would be reporting BAU/ml.

9. You refer to significant decreases in line 154, but I do not see any description of statistical analysis in any of the figure legends or inclusion of common symbols for p values (ns, *, **, ***, and ****) on the figures themselves. Appropriate statistical analysis should be performed and should be documented in the methods section as well as in the figure legends, with the symbols for the results included on the graphs. If you are comparing the D15, D60, and D90 values to the D0 value within a group I would suggest running multiple t-test (parametric) or Mann-Whitney (non-parametric) with an appropriate compensation for multiple comparisons as possible choices.

10. Make sure that you refer to ‘fold changes’ throughout. There are a few instances of referring to ‘times changes’. Consistency in terminology is extremely helpful to the reader.

11. It seems that the information in Table 2 (other than the Age and Gender) are the same data included in Figures 1, 2, and 3. Either method of presentation is acceptable (I happen to prefer the graphs), but double presentation of the same data should be avoided unless it is justified by applying a different type of analysis (as in Tables 3 and 4 where you break the dataset apart by gender).

12. The sentence in lines 187-190 is essentially duplicated in lines 190-193.

13. In the discussion, use a comma instead of a period to denote the thousands place when referring to IgG measurements (lines 228, 231, 239) since that is the convention you have used elsewhere in the paper.

14. Lines 231-235 are confusing, it appears there is an accidental sentence fragment that results in contradictory information.

15. I believe that the results reference in line 251 are referring to the percent neutralization, not the percent of neutralizing antibodies (which I take to mean what portion of the total antibody response is neutralizing)

16. As a general comment, you state that no new data were generated by this study, but you did indeed generate quite a bit of data. For 300 individuals, you have demographic and health data as well as multiple types of antibody measurements collected at multiple timepoints. I strongly suggest putting all of this data together in a supplemental file to accompany your manuscript.

Author Response

Dear Reviewer,

Thank you very much for your valuable comments and contribution. all comments have been carefully checked and edited on the manuscript. Following your suggestion, editing service was taken from native english speaker.

You can also see our response below.

Thank you, Regards

Assoc. Prof. Mehmet Demirci

Response to Reviewer 1:

The authors report the results of multiple antibody assays (anti-nucleocapsid IgG, anti-RBD IgG, and a surrogate neutralization assay reported as the percent inhibition of RBD binding to ACE2) performed at 0, 15, 60, and 90 days following a third dose of the BNT162b2 mRNA vaccine. The data was collected from a relatively large cohort of 300 individuals. Analysis focused on the impact of prior SARS-CoV-2 infection on the magnitude and duration of the vaccine-induced antibody response, as well as the impact of gender. They find that prior SARS-CoV-2 does significantly increase the amount of anti-RBD IgG at later timepoints, but that it doesn’t cause a corresponding increase in surrogate neutralizing antibody levels. The authors’ premise and data appear sound, but editing is required to ensure clarity for the reader.

  1. The definition of participants with and without prior SARS-CoV-2 infection is critical to the analysis and interpretation of the reported results. The methods section taken alone is not clear about whether that is defined as having anti-nucleocapsid antibodies or an affirmative answer to a survey result; this wasn’t made fully clear until the results in lines 146-150. I then got confused again when reading lines 163-167 – is this stating that 98 of the 154 women in the study (so, 63.6% of the women) were not previously infected as defined by the presence of N antibody, and 56/154 (so, 36.4% of the women) were previously infected? For the males (lines 179-182), shouldn’t it say 110/146=75.3% of men were not infected and 36/110=24.7% of men were not previously infected? Modify the methods section to more explicitly state that only the antibody data was used to define previously infected vs. no previous infection. Also, as the results of the survey question about previous SARS-CoV-2 infection do not appear to have been used in this study, it may be easiest to remove that text and refer only to data which was subsequently used for either generating or interpreting results or for inclusion/exclusion decisions.

Response: Thank you very much for your valuable comments and contribution. Following your suggestion, the evaluation criteria for previous SARS-COV infection have been clarified. The proportioning according to the total number was adjusted for male and female groups. The information that SARS-CoV-2 infection was questioned was removed.

  1. Regarding the exclusion criteria, the exclusion of individuals <18 years old is understandable due to the difficulty in obtaining truly informed consent from minors. However, please provide the rationale behind excluding those <50 years of age and/or presenting with any co-morbidities. Aren’t these the very groups at most risk from COVID-19, and wouldn’t data on this population be especially valuable? This could be included in the limitations section.

Response: Thank you very much for your valuable comments and contribution. Individuals aged 50 years and older were not included in our study because they were more physically inactive and had more comorbidities than the group included in the study. Following your suggestion, limitation section revised for this comment.

  1. There are a few instances where COVID-19 and SARS-CoV-2 are used interchangeably in the text and figures. Please do a close edit to make sure that it’s always COVID-19 when describing the disease and SARS-CoV-2 when referring to the virus. For example, line 57 should refer to the immune response against SARS-CoV-2 (the virus), not COVID-19 (the disease). Similarly, the labels of Figures 1, 2, and 3 refer to COVID-19 infection when they should technically refer to SARS-CoV-2 infection.

Response: Thank you very much for your valuable comments and contribution. Following your suggestion, sentences revised for the manuscript.

  1. Lines 45-46. I disagree somewhat is the phrasing here. Vaccination was able to help control the spread during the original and alpha variant waves, but whether it still decreases transmission in the face of the delta and omicron variants is far less certain due to the large number of breakthrough infections.

Response: Thank you very much for your valuable comments and contribution. Following your suggestion, this sentence revised for the manuscript.

  1. Change the wording regarding the surrogate neutralization assay. Clarify that this assay is measuring the presence of antibody that blocks the binding of RBD to ACE2, and that you are detecting potentially neutralizing or likely neutralizing antibodies, not neutralizing antibodies. The assay is fully valid, but make sure its clear throughout the paper that the data are for this surrogate assay, not a traditional neutralization assay by referring to the snAb that were actually measured, not nAb which was not measured.

Response: Thank you very much for your valuable comments and contribution. the sentence has been clarified and explanation has been added as you specified.

  1. Why were the %IH cutoffs of ≥35 for a positive and <20 for a negative chosen (lines 107-108), and how were samples that fell between those two cutoffs handled?

Response: Thank you very much for your valuable comments and contribution. the sentence has been clarified. Between those two cutoffs not found but according to manufacturer’s instruction. This result need to consider as borderline.

  1. A correction for multiple comparisons needs to be applied to your statistical analysis.

Response: Thank you very much for your valuable comments and contribution. Following your suggestion, Correction was made.

  1. Table 1 and Figure 2 refer to the anti-S-RBD IgG levels in terms of AU/ml. Is this correct? I assumed from the methods section that you would be reporting BAU/ml.

Response: Thank you very much for your valuable comments and contribution. Following your suggestion, this situation has been clarified. Results in BAU/ml are not presented.

  1. You refer to significant decreases in line 154, but I do not see any description of statistical analysis in any of the figure legends or inclusion of common symbols for p values (ns, *, **, ***, and ****) on the figures themselves. Appropriate statistical analysis should be performed and should be documented in the methods section as well as in the figure legends, with the symbols for the results included on the graphs. If you are comparing the D15, D60, and D90 values to the D0 value within a group I would suggest running multiple t-test (parametric) or Mann-Whitney (non-parametric) with an appropriate compensation for multiple comparisons as possible choices.

Response: Thank you very much for your valuable comments and contribution. Since the figures were taken directly from the SPSS programme, no additions could be made to these graphs.

  1. Make sure that you refer to ‘fold changes’ throughout. There are a few instances of referring to ‘times changes’. Consistency in terminology is extremely helpful to the reader.

Response: Thank you very much for your valuable comments and contribution. Following your suggestion, terminology were changed

  1. It seems that the information in Table 2 (other than the Age and Gender) are the same data included in Figures 1, 2, and 3. Either method of presentation is acceptable (I happen to prefer the graphs), but double presentation of the same data should be avoided unless it is justified by applying a different type of analysis (as in Tables 3 and 4 where you break the dataset apart by gender).

Response: Thank you very much for your valuable comments and contribution. p values are presented in the tables for illustration

  1. The sentence in lines 187-190 is essentially duplicated in lines 190-193.

Response: Thank you very much for your valuable comments and contribution. Following your suggestion, Correction was made.

  1. In the discussion, use a comma instead of a period to denote the thousands place when referring to IgG measurements (lines 228, 231, 239) since that is the convention you have used elsewhere in the paper.

Response: Thank you very much for your valuable comments and contribution. Following your suggestion, Correction was made.

  1. Lines 231-235 are confusing, it appears there is an accidental sentence fragment that results in contradictory information.

Response: Thank you very much for your valuable comments and contribution. Following your suggestion, Correction was made.

  1. I believe that the results reference in line 251 are referring to the percent neutralization, not the percent of neutralizing antibodies (which I take to mean what portion of the total antibody response is neutralizing)

Response: Thank you very much for your valuable comments and contribution. True

  1. As a general comment, you state that no new data were generated by this study, but you did indeed generate quite a bit of data. For 300 individuals, you have demographic and health data as well as multiple types of antibody measurements collected at multiple timepoints. I strongly suggest putting all of this data together in a supplemental file to accompany your manuscript.

Response: Thank you very much for your valuable comments and contribution. All data will be available upon request from the authors for new data generation. 

Reviewer 2 Report

The authors presented the manuscript on " Could prior COVID-19 affect the neutralizing antibody after the third BNT162b2 booster dose: A longitudinal study" . Similar studies have been carried out earlier and they provided similar evidence. In this study authors did not mention the side effects of third booster dose in study participants. Pre-exposure to different variants may have different effects. How the pre-exposure variant was determined. 

Author Response

Thank you very much for your valuable comments and contribution. all comments have been carefully checked and edited on the manuscript. Following your suggestion, editing service was taken from native english speaker.

You can also see our response below.

Thank you, Regards

Assoc. Prof. Mehmet Demirci

Response to Reviewer 2:

The authors presented the manuscript on " Could prior COVID-19 affect the neutralizing antibody after the third BNT162b2 booster dose: A longitudinal study" . Similar studies have been carried out earlier and they provided similar evidence. In this study authors did not mention the side effects of third booster dose in study participants. Pre-exposure to different variants may have different effects. How the pre-exposure variant was determined. 

Response: Thank you very much for your valuable comments and contribution. Data on side effects are kept and stored by the Ministry of Health. therefore, these data could not be monitored in this study.

Reviewer 3 Report

The authors report data from a study of 300 individuals on the antibody (SRBD and surrogate neutralizing antibodies (snAb)) after two dose vaccination with BNT162b2 before a third dose with the same vaccine.  The aim was to determine if an additional booster dose of the vaccine increased antibody levels to SARS- CoV2.  The study also determined that level and duration of these antibodies up to 90 days post vaccination.

The presented data basically indicates a small but not significant increase in antibody response after the third vaccination.  As expected, the antibody levels did show a slow decline 90 days after vaccination.  This data is somewhat redundant of other reports in the literature, albeit a different population of individuals were followed in the current study.  Individuals would be interested in the results as the data does shed some information on antibody production following multiple vaccination and duration of the antibody response.

The data does tend to indicate that multiple boosters seem to be beneficial  inducing a heightened antibody response.  The paper is well presented, there are a few grammatical corrections needed.

Author Response

Thank you very much for your valuable comments and contribution. all comments have been carefully checked and edited on the manuscript. Following your suggestion, editing service was taken from native english speaker.

You can also see our response below.

Thank you, Regards

Assoc. Prof. Mehmet Demirci

Response to Reviewer 3:

The authors report data from a study of 300 individuals on the antibody (SRBD and surrogate neutralizing antibodies (snAb)) after two dose vaccination with BNT162b2 before a third dose with the same vaccine.  The aim was to determine if an additional booster dose of the vaccine increased antibody levels to SARS- CoV2.  The study also determined that level and duration of these antibodies up to 90 days post vaccination.

The presented data basically indicates a small but not significant increase in antibody response after the third vaccination.  As expected, the antibody levels did show a slow decline 90 days after vaccination.  This data is somewhat redundant of other reports in the literature, albeit a different population of individuals were followed in the current study.  Individuals would be interested in the results as the data does shed some information on antibody production following multiple vaccination and duration of the antibody response.

The data does tend to indicate that multiple boosters seem to be beneficial  inducing a heightened antibody response.  The paper is well presented, there are a few grammatical corrections needed.

Response: Thank you very much for your valuable comments and contribution. Following your suggestion, editing service was taken from native english speaker.